# Perinatal complications of the maternal–Fetal dyad in primiparous women subjected to vaginal delivery versus elective cesarean section: A retrospective study of clinical results associated with bioethical precepts

Carlos Henrique Mascarenhas Silva[1,2]*, Cláudia Lourdes Soares Laranjeira[2,3], Carolina Soares Barros de Melo[2], Lorena Ventura Brandão[2‡], Gabriela Costa Oliveira[2‡], Augusto Henrique Fulgêncio Brandão[2,4‡], Rui Nunes[1]

1 Faculty of Medicine of the University of Porto, Porto, Portugal, 2 Obstetrics and Gynecology Unit, Mater Dei Health Network, Belo Horizonte, Brazil, 3 Department of Women's Health, School of Medical Sciences of Minas Gerais, Federal University of Minas Gerais, Belo Horizonte, Brazil, 4 Obstetrics and Gynecology Department, Federal University of Minas Gerais, Belo Horizonte, Brazil

☉ These authors contributed equally to this work.
‡ LVB, GCO and AHFB also contributed equally to this work.
* carloshenrique@materdei.com.br

## Abstract

The obstetrics field is undergoing transformation and committing to ensuring the autonomy of pregnant women in decisions related to birth based on scientific information. The physiological process of birth typically results in vaginal delivery, but medicine has evolved to include obstetric surgeries that are safe and result in few perioperative complications, especially when cesarean section is performed from 39 weeks of gestational age. Thus, the question is whether clinicians should interfere with pregnant women's freedom to choose their mode of delivery by trying to persuade them to choose vaginal delivery. The objective was to analyze the perinatal complications of the maternal–fetal dyad in primiparous women subjected to vaginal delivery versus elective cesarean section with respect to the bioethical precepts of autonomy, beneficence and nonmaleficence. In total, 2,507 women, including 1,807 (72.1%) with vaginal deliveries and 700 (27.9%) with cesarean deliveries, were analyzed between 2017 and 2020. There was no difference between the types of delivery in maternal readmission, death, admission to the intensive care unit, an Apgar score <7 in the 5th minute of life, maternal blood transfusion or comorbidities of the mothers or newborns. The elective cesarean section group showed less need for therapeutic uterotonics. In primigravidae, it was observed that elective cesarean section did not present a higher risk of complications than vaginal delivery. Therefore, this guarantees the autonomy and right of the individual to choose the mode of delivery.

**Data Availability Statement:** All relevant data are within the paper.

**Funding:** The author(s) received no specific funding for this work.

**Competing interests:** The authors have declared that no competing interests exist.

# Introduction

Although pregnancy and parturition are natural processes, negative or positive experiences during the postpartum and pregnancy periods can influence current or future pregnancies, triggering profound transformations [1]. Current studies based on scientific evidence have shown that the presence of the family in the environment, the active involvement of pregnant women in decision-making and their behaviors are able to improve the clinical outcomes of both the women and their newborns, positively enriching the experience of pregnancy [1, 2].

Autonomy is one of the four pillars of bioethics and is fundamental for humanized and respectful care during birth [3]. Obstetric care is undergoing a transformation process to ensure autonomy and places the parturient as the center of the entire process [4].

Shared responsibility is possible through dialog and trust with the care team from the beginning of prenatal care until the time of delivery and depends on cultural, social, emotional, and economic conditions, among others. For this, women need to be welcomed and properly informed to understand what choices should be made based on quality technical scientific knowledge together with their care team [5].

Vaginal delivery is often not considered by women due to fear of pain, fear of not being able to give birth, insecurity about their future sexual life and possible perineal complications. In addition, for vaginal delivery, women must be empowered and emotionally and physically prepared for the event, even though it is physiological [6, 7].

Insecurities and doubts should be addressed during obstetric care; however, even after being informed of all the risks and benefits, based on quality information, patients may remain determined to have an abdominal delivery. In this case, the choice of cesarean section should be respected, without prejudice or attempts at coercion [8].

Medical-hospital care in Brazil is struggling to achieve the statistical threshold recommended by the World Health Organization (WHO), which, in 1985, claimed that the mode of vaginal delivery was the most beneficial and advised that the percentage of cesarean sections should not exceed 15% of the total number of births [9]. Thereafter, vaginal delivery was imposed in maternity wards, even though medicine has advanced rapidly to the point of achieving safe surgeries with low rates of perinatal complications, especially for elective cesarean sections in pregnant women at usual risk [10]. This leads to questions regarding the extent to which the autonomy of pregnant women is respected in the choice of delivery route.

Currently, there are several protocols regarding the issue of on-demand cesarean section, and they advise its implementation starting at 39 weeks of gestation to reduce the rates of neonatal respiratory distress and improve morbidity and mortality, thereby making elective abdominal delivery safe in the medium and short term, at the level of vaginal delivery [11–13].

According to resolution number 2.144 in 2016 and the reaffirmation with new additions to resolution 2.284 in 2020, Brazil's Federal Council of Medicine advises that doctors adhere to patients' desires to undergo cesarean section as long as the pregnancy has completed a minimum of 39 weeks of gestation (273 days) and the decision is documented in medical records along with a signed informed consent form. This guarantees women's autonomy and adheres to the principle of nonmaleficence by ensuring fetal maturity to reduce neonatal complications [14, 15].

There are also national laws and projects in Brazil, such as PL 3.635/2019, that defend the right to elective cesarean section, respecting the autonomy of the patient and the care teams in both public and private services in the country [16].

We also have an important defining basis for the autonomy of patients regarding the choice of the mode of delivery that was discussed at the United Nations International Convention on the Civil and Political Rights of Women and included in the chapter on respect for the reproductive rights of women and in the scope of the nondiscrimination policy, associated with

article 1 of the Convention for the Elimination of Forms of Discrimination against Women (CEDAW), in 1979. This convention stipulates that "any distinction, exclusion or restriction on the basis of sex which has the effect or purpose of impairing or nullifying the recognition, enjoyment or exercise by a woman, irrespective of her marital status, on the basis of equality between men and women, of human rights or fundamental freedoms in the political, economic, social, cultural civil or any other field" should be avoided. Thus, ensuring the right of women to choose their mode of delivery is in absolute agreement with the proposals contained in the cited statements, since capable individuals have the right to consider their options and make their own choices, whether consenting to or refusing medical treatment [17].

Beneficence and nonmaleficence are two other concepts of bioethics that must be considered when health care professionals are dealing with patients and their options and desires. Doing good and not doing evil, direct concepts of these two precepts, must be continuously evaluated when analyzing the results obtained for the assistance provided. These concepts are independent and interdependent with autonomy [18].

Thus, the objective of this study was to compare the safety and perinatal results of mothers and newborns among primiparous women subjected to term vaginal delivery or elective on-demand cesarean section after 39 weeks of gestation in a private quaternary hospital in Belo Horizonte and to discuss the bioethical principles related to the results.

## Methodology

This was a retrospective observational study. Data were collected after approval by the ethics committee (CAAE 36135120.5.0000.5128, opinion 4,400,669).

The inclusion criteria were as follows: the presence of a freely signed informed consent form specific for obstetric care, according to the choice made at the time of hospitalization; primigravida status; a gestational age above 37 weeks for vaginal birth or 39 weeks for elective cesarean section; fetuses without malformations; and cephalic presentation.

The outcomes analyzed included gestational age, type of anesthesia, newborn weight, an Apgar score less than seven in the fifth minute of life, referral to room-in or admission to a neonatal or adult intensive care unit (ICU) in the first 24 hours of life, immediate breastfeeding, maternal and/or early neonatal death, maternal and/or neonatal readmission, increased uterine bleeding, the need for blood transfusion, comorbidities, the use of uterotonic drugs and maternal hospitalization time.

Categorical variables are presented as absolute and relative frequencies, and numerical variables are presented as the mean ± standard deviation and median (1st quartile-3rd quartile). The numerical variables were subjected to the Anderson–Darling normality test, and Student's t test and the Mann–Whitney test were used to compare means between groups. The associations between categorical variables were evaluated by the chi-square test (with a simulated p value in the case of expected frequencies less than 1).

A significance level of 1% was considered, and the analyses were performed using R version 4.0.3 software.

## Results

A total of 2,507 women admitted to a private quaternary-level hospital in the care of a team of gynecologists and obstetricians who followed the institutional conduct discussed in periodic clinical meetings with critical analysis between January 2017 and March 2020 were analyzed; 1,807 (72.1%) women had vaginal deliveries and 700 (27.9%) had elective cesarean sections.

In the Table 1, "N" represented in the characteristics column represents the valid number of each evaluated variable.

**Table 1. Comparison of characteristics and outcomes of women who had vaginal births and cesarean sections.**

| Characteristic | General (n = 2,507) | Vaginal delivery (n = 1,807) | On-demand Cesarean section (n = 700) | P value |
|---|---|---|---|---|
| **Maternal age** (years) (n = 1,515) | 30.6 ± 5.0 | 30.0 ± 4.9 | 32.1 ± 4.8 | <0.001$^M$ |
| | 31.0 (28.0–34.0) * | 30.0 (27.0–34.0) * | 32.0 (29.0–35.0) * | |
| **Accommodation** (n = 1,244) | | n = 692 | n = 552 | - - |
| Set | 1,244 (100%) | 692 (100%) | 552 (100%) | |
| ICU | 0 (0.0%) | 0 (0.0%) | 0 (0.0%) | |
| **Anesthesia type** (n = 2,253) | | n = 1,807 | n = 446 | <0.001$^Q$ |
| No | 12 (0.5%) | 12 (0.7%) | 0 (0.0%) | |
| Location | 3 (0.1%) | 3 (0.2%) | 0 (0.0%) | |
| Epidural | 1,779 (79.0%) | 1,777 (98.3%) | 2 (0.4%) | |
| Spinal anesthesia | 459 (20.4%) | 15 (0.8%) | 444 (99.6%) | |
| **Gestational age** (weeks) (n = 2,498) | 39.4 ± 1.0 | 39.3 ± 1.0 | 39.7 ± 0.8 | <0.001$^M$ |
| | 39.5 (38.7–40.1)* | 39.3 (38.6–40.0)* | 40.0 (39.1–40.1) * | |
| **Weight** (kg) (n = 2.485) | 3,213.1 ± 382.5 | 3,155.8 ± 365.3 | 3,360.3 ± 386.8 | <0.001$^T$ |
| | 3,210.0 (2,960.0–3,460.0) * | 3,160.0 (2,920.0–3,405.0) * | 3,350.0 (3,100.0–3,620.0) * | |
| **PICU admission** | | n = 1,807 | n = 700 | 0.043$^Q$ |
| Yes | 72 (2.9%) | 60 (3.3%) | 12 (1.7%) | |
| No | 2,435 (97.1%) | 1,747 (96.7%) | 688 (98.3%) | |
| **5-minute APGAR score** | | n = 1,807 | n = 700 | 0.063$^Q$ |
| < 7 | 18 (0.7%) | 17 (0.9%) | 1 (0.1%) | |
| ≥ 7 | 2,489 (99.3%) | 1,790 (99.1%) | 699 (99.9%) | |
| **Immediate breastfeeding** | | n = 1,807 | n = 700 | <0.001$^Q$ |
| Yes | 2,414 (96.3%) | 1,714 (94.9%) | 700 (100.0%) | |
| No | 93 (3.7%) | 93 (5.1%) | 0 (0.0%) | |
| **Maternal death** | | n = 1,807 | n = 700 | - - |
| Yes | 0 (0.0%) | 0 (0.0%) | 0 (0.0%) | |
| No | 2,507 (100.0%) | 1,807 (100.0%) | 700 (100.0%) | |
| **Neonatal death** | | n = 1,804 | n = 700 | - - |
| Yes | 0 (0.0%) | 0 (0.0%) | 0 (0.0%) | |
| No | 2,504 (100.0%) | 1,804 (100.0%) | 700 (100.0%) | |
| **Maternal readmission** | | n = 1,807 | n = 700 | 0.569$^Q$ |
| Yes | 32 (1.3%) | 25 (1.4%) | 7 (1.0%) | |
| No | 2,475 (98.7%) | 1,782 (98.6%) | 693 (99.0%) | |
| **Neonatal readmission** (n = 2.499) | | n = 1,799 | n = 700 | <0.001$^Q$ |
| Yes | 183 (7.3%) | 177 (9.8%) | 6 (0.9%) | |
| No | 2,316 (92.7%) | 1,622 (90.2%) | 694 (99.1%) | |
| **Uterine bleeding** | | n = 1,807 | n = 700 | <0.001$^Q$ |
| Usual | 2,222 (88.6%) | 1,574 (87.1%) | 648 (92.6%) | |
| Increased | 285 (11.4%) | 233 (12.9%) | 52 (7.4%) | |
| **Transfusion** | | n = 1,807 | n = 700 | >0.999$^Q$ |
| Yes | 7 (0.3%) | 5 (0.3%) | 2 (0.3%) | |
| No | 2,500 (99.7%) | 1,802 (99.7%) | 698 (99.7%) | |
| **Use of uterotonic drugs** (n = 2,506) | | n = 1,806 | n = 700 | <0.001$^Q$ |
| Prophylactic | 1,660 (66.2%) | 1,014 (56.1%) | 646 (92.3%) | |
| Therapeutic | 845 (33.7%) | 792 (43.9%) | 53 (7.6%) | |
| Puerperal hysterectomy | 1 (0.0%) | 0 (0.0%) | 1 (0.1%) | |
| **Comorbidities** (n = 2,496) | | n = 1,806 | n = 690 | 0.146$^Q$ |
| Yes | 564 (22.6%) | 394 (21.8%) | 170 (24.6%) | |

*(Continued)*

**Table 1.** (Continued)

| Characteristic | General (n = 2,507) | Vaginal delivery (n = 1,807) | On-demand Cesarean section (n = 700) | P value |
|---|---|---|---|---|
| No | 1,932 (77.4%) | 1,412 (78.2%) | (75.4%) | |

^M Mann–Whitney test; ^T Student's t test; ^Q Chi-square test

* Represents the median (1° quartile– 3° quartile)

The maternal age of the analyzed patients ranged from 27 to 35 years; no puerperal woman was referred to the ICU, and no maternal or neonatal death was documented in the analyzed sample.

There was no significant difference in admission to a pediatric intensive care unit (PICU), Apgar scores lower than seven in the fifth minute of life, maternal readmission, the need for blood transfusion or the percentage of comorbidities between the groups.

There was a difference in the mean gestational age between the vaginal delivery group (38 weeks + 6 days to 40 weeks) and the on-demand cesarean section group (39 weeks + 7 days to 40 weeks + 1 day).

The vaginal delivery group had a higher number of neonatal readmissions and higher rates of puerperal bleeding and consequently a greater need for the therapeutic use of uterotonics. Only one patient in the elective cesarean section group underwent puerperal hysterectomy due to the failure of the clinical treatment for puerperal hemorrhage.

The elective cesarean section group had a higher rate of immediate breastfeeding than the vaginal delivery group.

The length of hospital stay after vaginal delivery ranged from 24 to 48 hours, and that after elective cesarean section ranged from 48 to 72 hours.

In the vaginal delivery group, 47.4% of the patients had an intact perineum: 14.7% had first-degree lacerations, 35% had second-degree lacerations, 2.3% had third-degree lacerations, 0.3% had fourth-degree lacerations, and 0.25% had cervix lacerations. Episiotomy was performed in 76.2% of the women, as shown in Table 2.

Table 3 describes the Robson classification of the nulliparous women evaluated in each group. In the vaginal delivery group, labor was induced in 24.2% of the women, while 75.8% were in spontaneous labor when admitted. Cesarean section was performed after 39 weeks of

**Table 2. Other characteristics of the women with vaginal births.**

| Outcomes | Patient number/Percentage |
|---|---|
| **Perineal lesion** (n = 1,790) | |
| 1st degree | 264 (14.7%) |
| 2nd degree | 627 (35.0%) |
| 3rd degree | 42 (2.3%) |
| 4th degree | 5 (0.3%) |
| Neck | 4 (0.2%) |
| No | 848 (47.4%) |
| **Episiotomy** (n = 1,806) | |
| Yes | 1,376 (76.2%) |
| No | 430 (23.8%) |
| **Hospital stay (hours)**–(n = 1,805) | 41.6 ± 17.6 |
| | 48.0 (24.0–48.0%) |

**Table 3. Distribution of the Robson classification by type of delivery.**

| Robson Classification | General (n = 2,506) | Vaginal delivery (n = 1,807) | Cesarean section (n = 699) |
|---|---|---|---|
| Group 1 | 1,514 (60.4%) | 1,369 (75.8%) | 145 (20.7%) |
| Group 2 | 989 (39.5%) | 439 (24.2%) | 553 (78.8%) |
| Group 3 | 0 (0.0) | 0 (0.0) | 0 (0.0) |
| Group 4 | 0 (0.0) | 0 (0.0) | 0 (0.0) |

gestational age and before the onset of labor (on-demand cesarean section) in 78% of the patients.

## Discussion

The differentiating feature of the study was the selection of nulliparous women who underwent on-demand elective cesarean sections after 39 weeks of gestation, respecting their autonomy of choosing the mode of delivery. Most articles that showed worse outcomes in the maternal–fetal dyad did not include these women. Instead, the results were based on the total number of cesarean sections without exclusions of intrapartum emergency cesarean section, previous cesarean section, maternal or fetal comorbidity-related indications for the procedure or gestational age, which can increase maternal and neonatal morbidity and mortality compared to term spontaneous vaginal delivery [19, 20].

The present study did not aim to encourage or promote the practice of cesarean sections in an indiscriminate manner, maintaining the recommendation of the American College of Gynecology and Obstetrics (ACOG) to encourage, but not obligate, pregnant women to undergo vaginal delivery and to address future aspects such as the risks of subsequent surgeries for those who choose to have more children, placenta previa, placental accreta and other long-term obstetric risks [12].

The results allow only for the corroboration of respect for the autonomy of pregnant women, without judgment or criticism from the team, guaranteeing them the right of choice due to the absence of greater maternal or fetal perioperative risks in the short and medium term, which are still erroneously used as an argument in environments that persuade pregnant women to undergo vaginal delivery, without ensuring their autonomy during parturition.

Another notable point of the study is that ensuring the autonomy of pregnant women regarding the delivery route did not negatively impact the rates of cesarean section, since the total percentage of cesarean sections was only 27.5% of all births, even in a supplementary health system that respects the autonomy of women, also reflecting the beneficence aspect of bioethics.

Although the vaginal delivery group had more neonatal readmissions, the main reason was late neonatal jaundice. Perinatal jaundice is associated with several factors, such as the presence of gestational diabetes mellitus, preeclampsia, the time of umbilical cord clamping, late elimination of meconium, blood incompatibility and significant weight loss during the first week, and is not directly related to the type of delivery [21, 22].

The main comorbidities in decreasing order in the vaginal delivery group were hypothyroidism, preeclampsia that did not meet the severity criteria, controlled chronic hypertension (CCH), gestational diabetes mellitus controlled with diet and asthma. In the cesarean section group, the main comorbidities were hypothyroidism, asthma, gestational diabetes mellitus controlled with diet and gestational hypertension (GH).

The vaginal delivery group had a higher incidence of puerperal bleeding, although it is not possible to conclude that cesarean section is preventive because the highest rate of

preeclampsia and CCH in the vaginal delivery group, which are known risk factors for puerperal bleeding, was not considered. Moreover, the labor time and the incidence of induction in the vaginal delivery group were not analyzed because prolonged births are associated with uterine hypotonia, the main cause of increased bleeding [23].

The elective on-demand cesarean section group had a higher rate of breastfeeding in the first hour of life, which may be indirectly related to the higher incidence of puerperal hemorrhage in the vaginal delivery group because breastfeeding is not recommended during the protocol and use of uterotonics. This isolated finding agrees with the data found in the literature, which show a lower rate of early breastfeeding because cesarean section performed before the onset of labor interferes with the release of oxytocin and, consequently, lactogenesis, making it difficult to initiate breastfeeding [24].

Although elective cesarean section hinders early lactation, evidence suggests that it has no effect on the interruption of breastfeeding before 6 months of life and does not impact the occurrence of respiratory infections, atopy or overweight/obesity in the first year of life [25].

Liu X. et al. conducted a retrospective cohort in 2015, analyzing 66,226 nulliparous women who had spontaneous vaginal delivery between 2007 and 2013 and comparing the outcomes with a group of women subjected to elective cesarean section. There was no significant difference in the rates of admission to the ICU, puerperal hemorrhage, maternal infection, thromboembolism, organ injury or perinatal mortality. The neonates who were delivered vaginally had higher rates of birth injuries, neonatal infection, hypoxic ischemic encephalopathy, and meconium aspiration. Conversely, the elective cesarean section group showed a higher rate of respiratory distress syndrome, although the minimum gestational age of 39 weeks for abdominal delivery was not observed. The study corroborates the data found in the present study, allowing us to argue the safety of the elective cesarean procedure compared to vaginal delivery [26].

Among the limiting factors of the study are the sample size and heterogeneity, although the analyses were performed after adjustment, considering the proportions in each group. Another bias is due to the data being from only one medical center with all the available structures and trained staff; therefore, more multicenter studies with different realities are needed to strengthen the scientific evidence.

However, although it is not possible to state that elective on-demand cesarean section is protective, it can be concluded that it does not worsen maternal or fetal outcomes in the short or medium term, thus ensuring respect for the bioethical precepts of autonomy, beneficence, and nonmaleficence.

Considering the limitation of this study for being a retrospective observational, accomplished in a single center that explores just some of the short-term outcomes of elective C-section offered to primigravida women at 39 weeks gestational age compared vaginal delivery, it's possible only suggest that elective abdominal delivery in a controlled environment is as safe as vaginal delivery in primigravidae and, therefore, should be respected when this is a woman's choice. It is also important that obstetricians are the guarantor of the choices and desires of pregnant women in the context of maternal health care.

However, more multicenter studies are needed to evaluate the security and other short- and long-term outcomes.

## Author Contributions

**Data curation:** Lorena Ventura Brandão, Gabriela Costa Oliveira.

**Formal analysis:** Cláudia Lourdes Soares Laranjeira.

**Investigation:** Carlos Henrique Mascarenhas Silva.

**Methodology:** Carlos Henrique Mascarenhas Silva, Cláudia Lourdes Soares Laranjeira.

**Project administration:** Carlos Henrique Mascarenhas Silva.

**Resources:** Lorena Ventura Brandão, Gabriela Costa Oliveira.

**Supervision:** Augusto Henrique Fulgêncio Brandão, Rui Nunes.

**Writing – original draft:** Carolina Soares Barros de Melo.

**Writing – review & editing:** Augusto Henrique Fulgêncio Brandão, Rui Nunes.

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
