## [Decision Letter · Decision Letter 0]

8 Jan 2023

PONE-D-22-30934Autonomy of pregnant women in choosing the mode of delivery: A retrospective studyPLOS ONE

Dear Dr. Silva ,

Thank you for submitting your manuscript to PLOS ONE. After careful consideration, we feel that it has merit however does not fully meet PLOS ONE’s publication criteria as it currently stands. Therefore, we invite you to submit a revised version of the manuscript that addresses the points raised during the review process.

Kindly review and respond to the comments by the reviewers.==============================

We look forward to receiving your revised manuscript.

Kind regards,

Bilal Sulaiman

Academic Editor

PLOS ONE

Journal Requirements:

 "The author(s) received no specific funding for this work. The funders had no role in study design, data collection and analysis, decision to publish, or preparation of the manuscript."

Additional Editor Comments:

This manuscript is okay, however there are minor observations made by the reviewers that need to be considered for response.

Reviewers' comments:

Reviewer's Responses to Questions

**Comments to the Author**

1. Is the manuscript technically sound, and do the data support the conclusions?

Reviewer #1: Partly

Reviewer #2: Yes

2. Has the statistical analysis been performed appropriately and rigorously? 

Reviewer #1: Yes

Reviewer #2: Yes

3. Have the authors made all data underlying the findings in their manuscript fully available?

Reviewer #1: Yes

Reviewer #2: Yes

4. Is the manuscript presented in an intelligible fashion and written in standard English?

Reviewer #1: Yes

Reviewer #2: Yes

5. Review Comments to the Author

Reviewer #1: The study was conducted among primiparous women , with primigravidae as an inclusion criteria, so the conclusion should point that the choice of delivery route for women should be offered equally to all primiparous women among whom the study was conducted and not to all women.

Reviewer #2: Please see attached file

6. PLOS authors have the option to publish the peer review history of their article (what does this mean?). If published, this will include your full peer review and any attached files.

Reviewer #1: **Yes: **Dr Mohammed Usman

Reviewer #2: **Yes: **Ibrahim Umar

---

## [Author Response · Author response to Decision Letter 0]

7 Mar 2023

Dear academic editor and reviewers,

We come through this letter to thank you for the comments and suggestions the where made, which helps us to review this paper. That certainly qualify and improve our text.

We inform you. that the changes suggested by the academic editor and reviewers have been implemented.

Some clarifications ar provided below:

1- Files names and formatting have been fixed as those from Plos One templates

2- The information regarding the waiver of the informed consente trem by the ethics committee was included in the methodology.

3- Regarding the funding: the author received NO specific funding for this work.

4- The frist bibliographic referente that had been retracted was replaced by a more recent one.

5- As Suggested by reviewer 1#, the discussion and conclusion were chanced and we make more clear that our data includes only primiparous women in this study.

---

## [Editor Report · Decision Letter 1]

24 Apr 2023

PONE-D-22-30934R1Autonomy of pregnant women in choosing the mode of delivery: A retrospective studyPLOS ONE

Dear Dr. Silva,

Thank you for submitting your manuscript to PLOS ONE. After careful consideration, we feel that it has merit but does not fully meet PLOS ONE’s publication criteria as it currently stands. Therefore, we invite you to submit a revised version of the manuscript that addresses the points raised during the review process.

 The authors did not address some of the observations by the second reviewer.The article is supposed to address Autonomy of pregnant women in choosing the route of delivery as suggested by the title. But the objective and result showed no evidence of that. The result is more of maternofoetal outcomes (perhaps Beneficence and non-maleficence.)As it has been done for table 1, there should be a descriptive text of the main finding of table 2 and 3.In table 1, what do the figures 1.151, 1.244, 2.253, 2.498 and 2.485 under maternal age, accommodation, anaesthesia, gestational age and weight respectively stand for?The Discussion should be directed to respect for person and autonomy.==============================

We look forward to receiving your revised manuscript.

Kind regards,

Bilal Sulaiman

Academic Editor

PLOS ONE

Additional Editor Comments (if provided):

The authors have made some changes to the manuscript but the following observations should be addressed.

1. The authors did not address some of the observations made by the second reviewer.

2. The article is supposed to address Autonomy of pregnant women in choosing the route of delivery as suggested by the title. But the objective and result showed no evidence of that. The result is more of materno-

fetal outcomes (perhaps Beneficence and Non-maleficence.)

3. As it has been done for table 1, there should be a descriptive text of the main findings of table 2 and 3.

4. In table 1, what do the figures 1.151, 1.244, 2.253, 2.498 and 2.485 under maternal age, accommodation, anesthesia, gestational age and weight respectively stand for?

5. The Discussion should be directed to respect for person and autonomy.
---

## [Author Response · Author response to Decision Letter 1]

29 May 2023

Dear academic editor and reviewer(s):

With this letter, we would like to thank you for the comments and suggestions which helped revise our work and that certainly perfect and improve our text.

We inform you that the changes suggested by the academic editor and reviewer(s) have been implemented.

Some clarifications are provided below:

As the data allow us to discuss bioethical aspects beyond autonomy, such as beneficence and non-maleficence, we have chosen to change the title in order to make the message clearer and more compatible with the results presented. Therefore, the new title presented is “Bioethical principles in choosing the mode of birth: a retrospective study”.

Regarding the data questioned in Table 1: As these are retrospective data collected from electronic medical records, we were unable to obtain a complete record of all the analyzed variables of the 2507 patients. Thus, the highlighted numbers are the specific N of each evaluated variable. Example: Of the 2507 medical records, 1251 had the maternal age record, 1244 had the type of accommodation, 2253 described the use of analgesia and 2485 had the weight record.

As suggested, we have added the descriptive result of the data presented in Tables 2 and 3.

---

## [Decision Letter · Decision Letter 2]

10 Jul 2023

PONE-D-22-30934R2Bioethical principles in choosing the mode of birth: a retrospective studyPLOS ONE

Dear Dr. Mascarenhas Silva,

Thank you for submitting your manuscript to PLOS ONE. After careful consideration, we feel that it has merit but does not fully meet PLOS ONE’s publication criteria as it currently stands. Therefore, we invite you to submit a revised version of the manuscript that addresses the points raised during the review process.

Please respond to all reviewers comment one by one

We look forward to receiving your revised manuscript.

Kind regards,

Ahmed Mohamed Maged, MD

Academic Editor

PLOS ONE

Reviewers' comments:

Reviewer's Responses to Questions

**Comments to the Author**

1. If the authors have adequately addressed your comments raised in a previous round of review and you feel that this manuscript is now acceptable for publication, you may indicate that here to bypass the “Comments to the Author” section, enter your conflict of interest statement in the “Confidential to Editor” section, and submit your "Accept" recommendation.

Reviewer #3: (No Response)

Reviewer #4: (No Response)

Reviewer #5: (No Response)

2. Is the manuscript technically sound, and do the data support the conclusions?

Reviewer #3: Yes

Reviewer #4: Partly

Reviewer #5: No

3. Has the statistical analysis been performed appropriately and rigorously? 

Reviewer #3: (No Response)

Reviewer #4: Yes

Reviewer #5: N/A

4. Have the authors made all data underlying the findings in their manuscript fully available?

Reviewer #3: (No Response)

Reviewer #4: No

Reviewer #5: Yes

5. Is the manuscript presented in an intelligible fashion and written in standard English?

Reviewer #3: (No Response)

Reviewer #4: No

Reviewer #5: No

6. Review Comments to the Author

Reviewer #3: (No Response)

Reviewer #4: Main review comment:

The title and the narrative of the manuscript in the introduction, discussion, and conclusion sections focus on several of the bioethical principles in choosing the mode of delivery. However, the stated objective of the study: "is to compare the safety and perinatal results of the mother and newborn among primiparous women subjected to term vaginal delivery or elective on-demand cesarean section after 39 weeks of gestational age." While the authors do make links between the outcomes of modes of delivery and the bioethical principles in choosing the mode of delivery, it should be noted that the study objective stated by the authors and the study results are not in fact related to bioethical principles at all. If the main objective of the study is to address the safety and perinatal outcomes of on-demand elective cesarean section at one medical centre, then the manuscript title is misleading and some of the narrative in the discussion and conclusion sections should be revised to be more logically drawn based on the stated objective and the actual results, scope, and limitations of the study. If the intent of the authors is to also review the bioethical principles involved in choosing the mode of delivery as an additional study objective, then the manuscript should be revised accordingly to clearly state this objective.

Minor review comments:

1) Several of the revision and formatting changes to the manuscript word document (Manuscript (2).docx) that were suggested by the academic editor have not been implemented in the second revision of the manuscript. Please review these changes and implement accordingly.

2) Please clarify how the data underlying the results will be available. If the underlying data will be available as supplemental information, will these data be available as supplementary files along with the manuscript?

3) The claim made in the first paragraph under the discussion section about the methodologies of other articles should be substantiated by references or citations to the relevant literature: "The differentiating feature of the study is the selection of on-demand elective cesarean sections in nulliparous women after 39 weeks of gestational age. Most articles that show worse outcomes in the maternal–fetal binomial did not do this. Instead, their results are based on the total number of cesarean sections without exclusions of intrapartum emergency cesarean section, iterativity, maternal or fetal comorbidity-related indications for the procedure or gestational age, which can increase maternal and neonatal morbidity and mortality compared to term spontaneous vaginal delivery."

Reviewer #5: I cannot accept the manuscript for publication in Plos One journal. I doesn’t sound and very poor written.

7. PLOS authors have the option to publish the peer review history of their article (what does this mean?). If published, this will include your full peer review and any attached files.

Reviewer #3: **Yes: **Osama Youssef

Reviewer #4: No

Reviewer #5: No

---

## [Author Response · Author response to Decision Letter 2]

10 Aug 2023

Dear reviewers:

We would like to thank you for the comments and suggestions that helped in the revision of the study and that certainly improved our text. We incorporated the changes suggested by the academic editor and reviewers in the revised draft.

The following are some clarifications:

- The academic editor's recommendations in review 2 were incorporated into this draft.

- For many years, parturients were coerced into vaginal delivery using the allegation that cesarean section, even elective, meant more risk and higher rates of morbidity and mortality for the maternal–fetal binomial. However, most studies did not exclude emergency cesarean sections or cesarean sections indicated in high-risk pregnancies when comparing outcomes. Our results elucidate the safety of the two modes of delivery, which allows indirect discussion of the right to autonomy in choosing the mode of delivery, in addition to other bioethical principles. Since bioethical principles are not measured or presented in our formal results, we chose to modify the title of the text, the objective, and the conclusion, as suggested by reviewer 4. To avoid generating doubts about the study, we modified the title to: Perinatal complications of the maternal–fetal dyad in primiparous women subjected to vaginal delivery versus elective cesarean section: A retrospective study of clinical results associated with bioethical precepts

- Regarding the questions about the data in Table 1:

1- We do not have data on the ethnicities of all the patients because these data are not usually included in the medical records of our hospital, as Brazil is a country with high population miscegenation.

2- Because these retrospective data were collected from electronic medical records, we were unable to fully record all the variables analyzed in the 2507 patients. Thus, the highlighted numbers (N) are the valid numbers of each variable evaluated. For example, of the 2507 medical records, 1251 included the maternal age, 1244 the type of accommodation, 2253 described the use of analgesia, and 2485 the weight. As the meaning of the N value generated doubt, we added this information to the results.

3- Different statistical tests were used according to the characteristics of the variables: the quantitative/numeral variables were evaluated by the Anderson–Darling normality test, which analyzes the distribution pattern. According to its results, the T test was used when the numeral variables presented a normal distribution and the Mann‒Whitney test when they did not present a normal distribution. The chi-squared test was used to compare the qualitative variables.

4- The results of the variables maternal age, gestational age, and weight are presented as medians (first quartile, third quartile). Please see the additions to the legend of Table 1.

All tests were performed by a professional statistician.

- There was a difference in gestational age between the vaginal delivery and elective cesarean section, as most cesarean sections were performed near 39 weeks of gestational age, as required by law, and vaginal delivery is a spontaneous and less predictable event, but this fact did not affect the results or conclusions.

- We changed the title of Column 2 of Table 2 to “Patient number/Percentage”, as suggested.

- The English grammar and usage of the text was revised by American Journal Experts (attached certificate).

- As recommended, two references were added that corroborate the statements of the first paragraph of the Discussion.

We await the final answer and are willing to answer any questions you may have.

---

## [Decision Letter · Decision Letter 3]

8 Sep 2023

PONE-D-22-30934R3Perinatal complications of the maternal–fetal dyad in primiparous women subjected to vaginal delivery versus elective cesarean section: A retrospective study of clinical results associated with bioethical preceptsPLOS ONE

Dear Dr. Mascarenhas Silva,

Thank you for submitting your manuscript to PLOS ONE. After careful consideration, we feel that it has merit but does not fully meet PLOS ONE’s publication criteria as it currently stands. Therefore, we invite you to submit a revised version of the manuscript that addresses the points raised during the review process.

Please respond to all reviewers comments

We look forward to receiving your revised manuscript.

Kind regards,

Ahmed Mohamed Maged, MD

Academic Editor

PLOS ONE

Journal Requirements:

Reviewers' comments:

Reviewer's Responses to Questions

**Comments to the Author**

1. If the authors have adequately addressed your comments raised in a previous round of review and you feel that this manuscript is now acceptable for publication, you may indicate that here to bypass the “Comments to the Author” section, enter your conflict of interest statement in the “Confidential to Editor” section, and submit your "Accept" recommendation.

Reviewer #3: All comments have been addressed

Reviewer #4: (No Response)

2. Is the manuscript technically sound, and do the data support the conclusions?

Reviewer #3: Yes

Reviewer #4: Yes

3. Has the statistical analysis been performed appropriately and rigorously? 

Reviewer #3: Yes

Reviewer #4: Yes

4. Have the authors made all data underlying the findings in their manuscript fully available?

Reviewer #3: Yes

Reviewer #4: Yes

5. Is the manuscript presented in an intelligible fashion and written in standard English?

Reviewer #3: Yes

Reviewer #4: Yes

6. Review Comments to the Author

Reviewer #3: I only have a minor suggestion in one of your tables that can make the readers easily follow it. I attached the suggestion.

Reviewer #4: This is a single centre retrospective observational study that explores some of the outcomes of elective C-section offered to primigravida women at 39 weeks gestational age compared vaginal delivery. While the authors observe that there no major differences between the cohorts with respect to "maternal readmission, death, admission to the intensive care unit, an Apgar score <7 in the 5th minute of life, maternal blood transfusion or comorbidities of the mothers or newborns," the conclusions drawn from this study are somewhat overstated and generalised. Longer-term clinical outcomes are difficult to ascertain with this kind of retrospective study design to definitively conclude that one mode of delivery in this patient population is as safe as another. I would encourage the authors to consider revising their conclusions to accurately reflect their findings specific to the patient population studied, and within the scope and limitations of this study design.

7. PLOS authors have the option to publish the peer review history of their article (what does this mean?). If published, this will include your full peer review and any attached files.

Reviewer #3: **Yes: **Osama Youssef, PhD

Reviewer #4: No

---

## [Author Response · Author response to Decision Letter 3]

28 Sep 2023

Dear academic editor and reviewers:

We would like to thank you for the comments and suggestions that helped in the revision of the study and that certainly improved our text. We incorporated the changes suggested by the academic editor and reviewers in the revised draft.

The following are some clarifications:

- The reviewer 3 recommendations were incorporated into table 1.

- The limitations of study are added in the conclusion as reviewer 4 suggest.

- References were review, there’s no retracted reference in this article. 

We await the final answer and are willing to answer any questions you may have.

---

## [Editor Report · Decision Letter 4]

2 Oct 2023

Perinatal complications of the maternal–fetal dyad in primiparous women subjected to vaginal delivery versus elective cesarean section: A retrospective study of clinical results associated with bioethical precepts

PONE-D-22-30934R4

Dear Dr. Mascarenhas Silva,

We’re pleased to inform you that your manuscript has been judged scientifically suitable for publication and will be formally accepted for publication once it meets all outstanding technical requirements.

Kind regards,

Ahmed Mohamed Maged, MD

Academic Editor

PLOS ONE
---

## [Editor Report · Acceptance letter]

5 Oct 2023

PONE-D-22-30934R4 

Perinatal complications of the maternal–fetal dyad in primiparous women subjected to vaginal delivery versus elective cesarean section: A retrospective study of clinical results associated with bioethical precepts 

Dear Dr. Mascarenhas Silva:

I'm pleased to inform you that your manuscript has been deemed suitable for publication in PLOS ONE. Congratulations! Your manuscript is now with our production department. 

Kind regards, 

on behalf of

Professor Ahmed Mohamed Maged 

Academic Editor

PLOS ONE